# Towards Codec-LM Co-design for Neural Codec Language Models

## Abstract

Neural codec language models (or *codec LMs*) are emerging as a powerful framework for text-to-speech (TTS) and other audio generation tasks. These models leverage advancements in language modeling and high-fidelity residual vector quantization (RVQ)-based audio codecs, which compress continuous waveforms into discrete codes for LMs to process. Despite the close interdependence of codecs and LMs in these systems, research on codecs and LMs has largely remained siloed. In this work, we bridge this gap by proposing several codec-LM co-design strategies, analyzing their effects on end-to-end TTS performance and efficiency. Specifically, we introduce three complementary techniques: (i) a *frame-wise codec encoder* that improves both LM log-likelihood and end-to-end TTS metrics, (ii) *LM codebook level dropout*, a method to efficiently navigate a portion of the codec-LM design space by training a single LM, and (iii) *increased codec frame duration*, which we show can accelerate inference while maintaining end-to-end performance. Our experiments demonstrate that combining all three co-design techniques results in doubled inference speed, and improvements in intelligibility, audio quality, and speaker control in TTS relative to a siloed baseline.

## 1 Introduction

Neural codec language models (or codec LMs) (van den Oord et al., 2017; Wu et al., 2024) have recently emerged as a prominent framework for text-to-speech (TTS) (Tan et al., 2021; Wang et al., 2023; Yang et al., 2024) and general audio generation tasks (van den Oord et al., 2016; Copet et al., 2023; Borsos et al., 2023; Yang et al., 2024), replacing autoregressive methods that model continuous raw waveforms (van den Oord et al., 2016; Kalchbrenner et al., 2018; Goel et al., 2022). The success of codec LMs can be attributed to improvements in the architecture, scaling, and efficiency of language models (LMs) (Vaswani et al., 2017; Brown et al., 2020; Dao et al., 2022; Gu & Dao, 2023), as well as increasingly high-fidelity convolutional audio codecs that employ the residual vector quantization (RVQ) technique (Zeghidour et al., 2021; Défossez et al., 2023; Kumar et al., 2023), bridging continuous-domain audio generation tasks with LM methods that model discrete tokens.

A codec LM-based TTS or audio generation system consists of two separately trained components working together: (i) a neural codec that is trained as an autoencoder to encode raw audio waveforms into discrete code sequences and to decode (reconstruct) the waveform from those codes, and (ii) an LM that models the code sequences autoregressively. Although these two components are closely coupled, they represent relatively isolated research areas. Research on codecs (Zeghidour et al., 2021; Défossez et al., 2023; Kumar et al., 2023; Ahn et al., 2024) primarily focuses on achieving higher compression rates (i.e., lower bandwidths) while maintaining reconstruction quality, rather than optimizing for downstream language modeling. Conversely, research on codec-based LMs typically treats the codec as a fixed module and explores how to best model the codec tokens, enhancing aspects such as conditioning (Borsos et al., 2023; Yang et al., 2024), RVQ code patterning (Copet et al., 2023; Yang et al., 2024), or non-autoregression (Wang et al., 2023). While the design space of codecs and LMs combined is too large to explore exhaustively, considering each in isolation may be suboptimal when the goal is to improve the end-to-end performance.

In this work, we aim to break the isolation and uncover co-design principles between the codec and the LM. We identify several aspects that play a key role in the interactions between the two, and substantially impact the end-to-end generation quality and/or efficiency. Leveraging these co-design

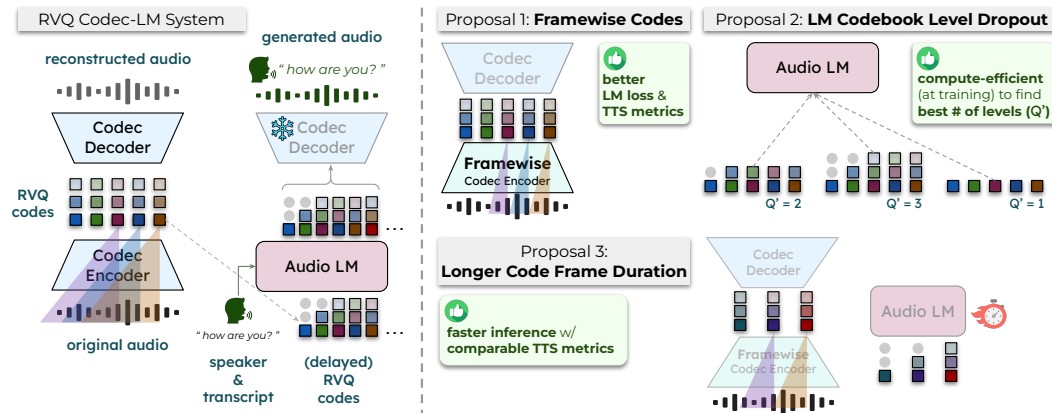

Figure 1: Overview of an RVQ-based codec-LM system for TTS (left), our contributions (right, **Proposals 1, 2 & 3**), and associated benefits. (Shaded triangles are receptive fields per code frame.)

insights, we propose actionable interventions which can improve the performance and efficiency (both at training and inference) of end-to-end audio generation systems.

First, we consider the receptive fields of neighboring RVQ code frames. In the context of RVQ audio codecs, we define a *frame* as a chunk of waveform samples (e.g., 512 samples) that correspond in time to one or more discrete *codes* (collectively, a *code frame*). A code frame contains multiple RVQ *levels* capturing increasingly fine-grained information. While typically, the convolutional receptive fields of neighboring code frames overlap both in the codec's encoder and decoder, the overlapping has different implications on the two seemingly symmetrical sides. In the decoder, it widens the information bottleneck enabling higher-fidelity reconstruction. In the encoder however, it may muddy the relationship between time in the waveform and time in the code frames, which we show can be detrimental to the downstream LM. To address this, we propose a *framewise* codec encoder, i.e., one which encodes each waveform frame independently. Our experiments show that using a framewise codec encoder (i) improves the downstream LM's log-likelihood (>8% higher), and (ii) leads to consistently better end-to-end metrics on both TTS and unconditional music generation tasks.

Second, given a trained RVQ codec, we propose a method for efficiently determining the subset of RVQ levels to use in LM training to improve end-to-end metrics. The residual structure of RVQ enables the LM to be trained on subsets of RVQ levels, raising the question of what number of levels to model for optimal end-to-end performance. Modeling more levels might presumably improve end-to-end audio generation performance by improving audio reconstruction quality. However, we observe in pilot experiments that increasing the number of levels past a certain point actually degrades end-to-end performance, possibly because the LM cannot effectively model such fine-grained information. Training multiple LMs to determine the optimal level account is a computationally expensive endeavor. Accordingly, we present *LM codebook level dropout*, a technique that trains a single LM on all possible level counts. We demonstrate that end-to-end performance results from a single LM training run using our technique on 12-level RVQ codes track the results of 12 LMs trained on each possible level count for the same number of gradient steps. Accordingly, our proposed technique dramatically improves the efficiency by which researchers can tune this important hyperparameter.

Finally, in addition to the number of levels, there exist two other salient codec hyperparameters, i.e., the frame duration and codebook size. In siloed codec design, these hyperparameters can be freely traded off with one another without affecting the codec's bitrate or audio reconstruction quality. However, in a co-design setting, these individual hyperparameters have unique interactions with the downstream LM. Crucially, LM sequence length is inversely proportional to frame duration, providing a strong motivation for using longer frames to increase the efficiency of the LM. We explore the end-to-end effects of increasing the frame duration while suitably adjusting the other two hyperparameters to maintain codec reconstruction quality. We show that doubling the frame duration of the default DAC codec (Kumar et al., 2023) to 22 milliseconds can result in doubled end-to-end inference speeds and comparable end-to-end metrics.

We base our experiments on a streamable (i.e., causal) variant of the DAC codec (Kumar et al., 2023), implementing our changes (i.e., framewise encoder, longer frame duration, and larger codebook size) without altering its architecture. We then train *Delay*-pattern LMs (Copet et al., 2023) for TTS, where LM codebook level dropout is applied, on the RVQ codes from our codecs.

Our contributions can be summarized as follows:

- We introduce a **framewise codec encoder** (Sec. 4.1), which leads to significant improvements in LM log-likelihood (>8% higher), and also consistently better end-to-end TTS metrics on intelligibility, audio quality, and speaker control (Table 1).

- We propose **LM codebook level dropout** (Sec. 4.2), which allows practitioners to efficiently tune a salient hyperparameter of the codec-LM design space in a single LM training run (Fig. 3).

- We show that using **longer frame durations** (Sec. 4.3) can increase end-to-end TTS inference speeds while preserving the end-to-end TTS metrics (Table 2).

- We demonstrate that combining all three co-design techniques doubles the end-to-end TTS inference speeds while *improving* all end-to-end TTS metrics (Table 3).

We plan to open source the implementation and pretrained weights of our framewise codec encoder, as well as the training-time procedure for LM codebook level dropout upon paper publication.

## 2 RELATED WORK

**Neural audio codecs.** Compressing and quantizing long, continuous audio waveforms into shorter discrete codes using a convolutional autoencoder was first proposed by van den Oord et al. (2017). Their proposed VQ-VAE method involves online K-Means for quantizing latent representations and a reconstruction objective on the decoder's output. Later, SoundStream (Zeghidour et al., 2021) introduced the 2D-structured Residual Vector Quantization (RVQ) to such codecs. This work also integrated a mixture of discriminators, a technique adoped from GAN-based audio synthesis (Goodfellow et al., 2014; Donahue et al., 2019; Kumar et al., 2019; Kong et al., 2020), on top of the decoder to enhance the perceptual quality of reconstructed waveforms—this RVQ-GAN setup has since been a norm for neural audio codecs. EnCodec (Défossez et al., 2023) and DAC (Kumar et al., 2023) further advanced the RVQ-GAN architecture with optimized discriminator setup, activation function, and (low) latent dimensionality. HILCodec (Ahn et al., 2024) showed that layer-wise variance constraining helps with the depth scaling of lightweight RVQ-GAN codecs. Overall, research in neural audio codecs has focused on achieving higher compression (i.e., lower bitrates) while maintaining audio reconstruction quality, rather than downstream audio generation, and often involved detailed architectural designs and tuning. In contrast, our work approaches codec design from an end-to-end audio generation practitioners' perspective, exploring codec hyperparameters that are both easily configurable and influential to the end-to-end system.

**LM-based end-to-end audio generation.** Autoregressive modeling of compressed discrete codes for audio waveforms was first proposed alongside VQ-VAE (van den Oord et al., 2017). AudioLM (Borsos et al., 2023) introduced a hierarchical LM approach that first generates *semantic tokens* (Hsu et al., 2021; Chung et al., 2021), derived from BERT-like pretraining (Devlin et al., 2019) on audio data, followed by RVQ codes (or *acoustic tokens*), resulting in better long-term coherence in generated audios. To navigate the efficiency-quality tradeoff given an RVQ codec, VALL-E (Wang et al., 2023) proposed non-autoregressive modeling for all RVQ levels except the coarsest one, and MusicGen (Copet et al., 2023) introduced the *Delay* pattern, dramatically shortening the sequence length while preserving key autoregressive dependencies. UniAudio (Yang et al., 2024) unified tokenization schemes for text, phonemes, audio, and symbolic music to build an LM for a wide range of audio generation tasks. Despite these advancements, all aforementioned work treated the audio codec, which is upstream from the LM, as a fixed component, leaving out the potential gains from a co-design between the codec and the LM.

**Co-design of audio codecs and LMs.** Compared to the two previously discussed areas, designing codecs with the goal of improving end-to-end audio generations is a relatively nascent direction. SpeechTokenizer (Zhang et al., 2024) proposed to distill information in semantic tokens (Hsu et al., 2021) into the first (coarsest) level of the RVQ codec, alleviating the need of using two

LMs (Borsos et al., 2023; Agostinelli et al., 2023) in tandem for semantic and acoustic RVQ tokens. Moshi (Défossez et al., 2024), a work conducted concurrently with ours, adopted this technique and used a causal codec setup to enable low-latency, streamable real-time voice conversations. Language-Codec (Ji et al., 2024) proposed to arrange the RVQ levels in a first-parallel, then-sequential fashion to distribute information more evenly among the RVQ levels. While the methods above improved the latency and/or quality of end-to-end generations, they focused on single, and highly specific, modifications to the codec. Meanwhile, our work investigate the downstream impact of multiple general RVQ codec hyperparameters in combination, painting a more complete picture for end-to-end system practitioners.

## 3  TECHNICAL BACKGROUND

**Residual vector quantization (RVQ)-based audio codecs.**  An RVQ-based audio codec compresses a continuous *waveform* $\boldsymbol{w} \in \mathbb{R}^{\text{Tf}_\text{s}}$, where T is the duration (in seconds) and $\text{f}_\text{s}$ is the sampling rate (in Hz) of the waveform, into discrete *codes* $\boldsymbol{x} \in \mathcal{V}^{\text{Tf}_\text{x} \times Q}$. Here $\mathcal{V} := \{1, 2, \ldots, |\mathcal{V}|\}$ represents the *codebook*, $\text{f}_\text{x}$ (typically much smaller than $\text{f}_\text{s}$) is the *frame rate* (in Hz) of the codec, and $Q$ is the number of *codebook levels* used to represent each frame. We also call downsampling rate of the codec, i.e., $\text{f}_\text{s}/\text{f}_\text{x}$, the *frame size* (an integer number of audio *samples*) and $1/\text{f}_\text{x}$ the *frame duration* (in seconds). The term *residual* refers to how the $Q$ codebook levels are structured to progressively refine the quantization (Zeghidour et al., 2021). Let the unquantized representation (i.e., the codec encoder output) for the $i$-th frame be denoted by $\boldsymbol{h}_i^{(1)} \in \mathbb{R}^D$, where $D$ is the codec encoder's output dimension. The RVQ process works iteratively for each level $q \in \{1, \ldots, Q\}$ on a frame-by-frame basis, quantizing the residual information from preceding levels using a level-wise learned codebook $\mathcal{C}^{(q)} : \mathcal{V} \to \mathbb{R}^D$. The operations at each level are:

$$x_{i,q} := \arg\min_{\tilde{x} \in \mathcal{V}} \|\boldsymbol{h}_i^{(q)} - \mathcal{C}^{(q)}(\tilde{x})\|_2^2 \tag{1}$$

$$\boldsymbol{h}_i^{(q+1)} := \boldsymbol{h}_i^{(q)} - \mathcal{C}^{(q)}(x_{i,q}), \tag{2}$$

where $x_{i,q} \in \mathcal{V}$ becomes an element in the code sequence $\boldsymbol{x}$, and $\mathcal{C}^{(q)}(x_{i,q}) \in \mathbb{R}^D$ is the quantized representation corresponding to $x_{i,q}$. The level-wise quantized representations are summed frame-by-frame, i.e., $\sum_q^Q \mathcal{C}^{(q)}(x_{i,q}); \ \forall i \in \{1, \ldots, \text{Tf}_\text{x}\}$, to produce the input to the codec decoder for reconstructing the original waveform.

The basic building blocks of an RVQ codec are 1D convolutional layers. Typically, these layers use symmetric padding, which introduces a dependency on future inputs, making the codec *non-causal* and unsuitable for low-latency streaming. To construct a *causal* codec, we can shift all paddings to the left of each layer's input (Défossez et al., 2023), which limits the theoretical latency to $1/\text{f}_\text{x}$, i.e., the frame duration. However, this causal setting slightly degrades audio reconstruction quality due to the lack of future context. In our work, we adopt the causal setting, which is also explored in concurrent work (Défossez et al., 2024), as low-latency streamability is critical in real-time applications.

The training process of an RVQ codec often includes *quantizer dropout* (Zeghidour et al., 2021; Kumar et al., 2023), which sometimes performs Eqn. (1) and Eqn. (2) for $Q_{\text{trunc}} < Q$ levels, forcing the codec to pack information in the lower codebook levels. Quantizer dropout enables the codec to encode and reconstruct audio waveforms at all $Q$ possible rates of compression, enhancing its versatility, and serves as a foundation for our codebook level dropout technique during LM training.

**Language modeling with *Delay* pattern of RVQ codes.**  Once the RVQ audio codec is trained, the remaining step in constructing an end-to-end audio generative model is to train an autoregressive LM on the discrete codes $\boldsymbol{x} \in \mathcal{V}^{\text{Tf}_\text{x} \times Q'}$, where $Q' \in \{1, \ldots, Q\}$ is a subset of the RVQ levels to model. A naive method to train LMs on these 2D-structured codes is to *flatten* them into a 1D sequence $\boldsymbol{x}^{(\text{flatten})} := [x_{1,1}, \ldots, x_{1,Q'}, x_{2,1}, \ldots, x_{\text{Tf}_\text{x}, Q'}]$. However, this would be highly inefficient as the LM's sequence length scales with $\text{T} \times Q'$. The *Delay* pattern proposed in (Copet et al., 2023) makes a good tradeoff between the efficiency and efficacy of modeling the RVQ codes $\boldsymbol{x}$. Instead of flattening, it shifts the $q$-th level of $\boldsymbol{x}$ to the right by $q$ positions, creating a shifted code sequence $\boldsymbol{x}^{(\text{delay})} \in \mathcal{V}^{(\text{Tf}_\text{x} + Q' - 1) \times Q'}$, where each frame (i.e., $\boldsymbol{x}_t^{(\text{delay})} \in \mathcal{V}^{Q'}$) can be specified by

$\boldsymbol{x}_t^{(\text{delay})} := [x_{t-q+1, q}]_{q=1}^{Q'}$. Then, an LM is trained to model:

$$p(\boldsymbol{x}) = p(\boldsymbol{x}^{(\text{delay})}) := \prod_{t=1}^{\text{Tf}_\text{x}+Q'-1} p(\boldsymbol{x}_t^{(\text{delay})} \mid \boldsymbol{x}_{<t}^{(\text{delay})}), \tag{3}$$

predicting the elements in each frame $\boldsymbol{x}_t^{(\text{delay})}$ in parallel. The Delay pattern has two desirable properties: (i) As the $Q'$ levels of a frame are collapsed into one LM timestep, the sequence length of the LM scales only with audio duration T rather than $\text{T} \times Q'$; (ii) Although some conditional independence is assumed between elements in $\boldsymbol{x}$, key dependencies like neighbor frames of the same level (e.g., $[x_{t-1,q}, x_{t,q}, x_{t+1,q}]$) and all levels of the same frame (e.g., $[x_{t,1}, x_{t,2}, \ldots, x_{t,Q'}]$) are still modeled sequentially as in the case where $\boldsymbol{x}$ is flattened and modeled autoregressively.

Though omitted in Eqn. (3), the LM is typically trained with conditions $\boldsymbol{y}$ expected from the user, e.g., text transcripts and speaker characteristics. Bringing all components together, our codec-LM audio generation system models the following distribution:

$$p(\boldsymbol{w}, \boldsymbol{x} \mid \boldsymbol{y}) := \underbrace{p(\boldsymbol{w} \mid \boldsymbol{x})}_{\text{learned by codec}} \cdot \underbrace{p(\boldsymbol{x} \mid \boldsymbol{y})}_{\text{learned by LM}}, \tag{4}$$

where conditional independence between waveform $\boldsymbol{w}$ and user inputs $\boldsymbol{y}$ is assumed given codes $\boldsymbol{x}$. We note that, in practice, $p(\boldsymbol{w} \mid \boldsymbol{x})$ is typically a deterministic mapping parameterized by the RVQ codec decoder.

## 4 METHOD

### 4.1 CODES WITH NON-OVERLAPPING RECEPTIVE FIELDS (**FRAMEWISE CODEC ENCODER**)

Most common RVQ audio codecs (Zeghidour et al., 2021; Défossez et al., 2023; Kumar et al., 2023) set the stride size of each 1D convolutional layer to be smaller than the filter size. This way the neighboring outputs (along the time dimension) have overlapping receptive fields, promoting the smoothness of transitions. When we consider the entire codec encoder, where multiple convolutional layers are stacked and gradually increase the receptive field of each intermediate representations, this overlapping property at each layer causes the receptive field of each code frame $\boldsymbol{x}_t$ to overlap with those of preceding code frames $[\boldsymbol{x}_{t-k}, \ldots, \boldsymbol{x}_{t-1}]$.[1][2] A similar property also holds in the codec decoder, i.e., each sample in the reconstructed waveform $\hat{\boldsymbol{w}}$ is influenced by multiple code frames.

If we reason about the frame-level overlaps, it is intuitive that they benefit the decoder, as the mutual information between multiple code frames can be leveraged for improved reconstruction. On the other hand, whether these overlaps are advantageous on the encoder side is less clear. They may provide the opportunity for the codec to pack information in high-complexity waveform segments (e.g., fast speech with frequent intonation changes) into neighboring code frames corresponding to low-information segments (e.g., silence), hence improving audio reconstruction. However, this could be detrimental for the downstream LM as each code frame may hold varying amounts of (confounding) information from preceding frames.

Therefore, we propose a setup where the codes are encoded *framewise*, i.e., each code frame $\boldsymbol{x}_t$ has a receptive field covering only $\text{f}_\text{s}/\text{f}_\text{x}$ waveform samples, without overlapping with other code frames. Operationally, this is achieved by reshaping the waveform (i.e., the inital input to the codec encoder) from $(B, \text{Tf}_\text{s}, 1)$, where the dimensions represent (batch, sequence, channels), to $(B\text{Tf}_\text{x}, \text{f}_\text{s}/\text{f}_\text{x}, 1)$. Since the downsampling rate of the entire encoder is precisely $\text{f}_\text{s}/\text{f}_\text{x}$, the final encoder output is of shape $(B\text{Tf}_\text{x}, 1, D)$, which we then reshape back to $(B, \text{Tf}_\text{x}, D)$ before quantization as in normal codecs with frame-level overlaps. Note that no architectural changes are required.[3]

This setup with *encoder-framewise* and *decoder-overlapping* receptive fields retains desirable properties such as leveraging mutual information between code frames for reconstruction, and smooth

---

[1]Here, we assume the codec is causal, and hence has no future dependencies.

[2]For example, for the architecture of DAC (Kumar et al., 2023), the extent of overlap is $k = 8$.

[3]We can also construct a framewise codec decoder using a similar reshaping operation on the (quantized) code frames, but we show (see Table 1) that it harms both reconstruction and end-to-end generation.

transitions of encoder intermediate representations, since their receptive fields remain overlapping within each frame. Also, the information unique to each frame of waveform samples is forced to be encoded *distinctly* into one code frame, instead of spilling over multiple code frames, which we anticipate might benefit downstream language modeling.

## 4.2 LM CODEBOOK LEVEL DROPOUT (**CL DROP**)

Here we propose a novel method designed to increase the efficiency of hyperparameter tuning for the number of codec RVQ levels $Q'$ used when training the downstream LM. The residual structure of RVQ quantization combined with quantizer dropout during codec training offers a useful property: an RVQ codec pre-trained with $Q$ levels can operate at inference time with any of $Q' \in \{1, \dots, Q\}$ code levels. Accordingly, this hyperparameter offers convenient flexibility for LM co-design as LMs with different $Q'$ can be trained without re-training the codec.

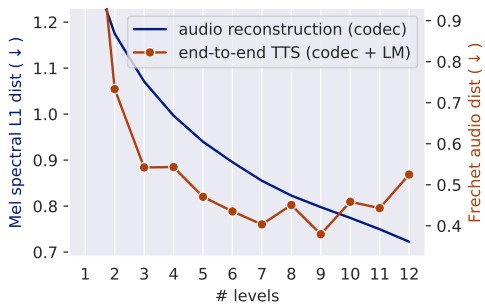

Figure 2: Impacts of # of codebook levels $Q'$ are different on codec-only *audio reconstruction* vs. *end-to-end TTS* involving both the codec and the LM. (frame duration $1/f_x = 11$ms; codebook size $|\mathcal{V}| = 2^{15}$.)

The choice of the hyperparameter $Q'$ can have a substantial impact on the end-to-end audio generation performance of the codec LM system. While increasing $Q'$ monotonically improves codec audio reconstruction due to a wider information bottleneck, its impact on the combined codec LM system is more ambiguous. Using too low of a $Q'$ value in the LM could result in poor audio quality, while using too high of a value could be detrimental as modeling finer-grained levels may: (i) present information that is too stochastic for the LM to process effectively, or (ii) shift the LM's capacity away from the coarser-grained levels which contain more crucial structural or semantic information about the audio.

To demonstrate this, we train a single RVQ codec on speech data with $Q = 12$ levels and train 12 LMs for text-to-speech (TTS) using each possible value of $Q' \in \{1, \dots, 12\}$. In Fig. 2, we first plot the codec audio reconstruction performance as measured by Mel-spectral L1 distance. We also plot the end-to-end codec LM system performance as measured by Fréchet audio distance (FAD) (Kilgour et al., 2019), an end-to-end metric for audio generation. We observe that end-to-end performance improves as the number of levels increases towards a global minima at 9 levels and deteriorates afterwards, as opposed to the monotonically improving curve of audio reconstruction.

Fig. 2 clearly demonstrates the potential importance of tuning $Q'$ in the codec LM. However, because LM training typically exceeds codec training in terms of required compute, the naive strategy of training $\mathcal{O}(Q')$ LMs to tune $Q'$ nullifies the potential compute savings of only needing to train a single codec. Therefore, we propose *codebook level dropout* (CL drop), which trains just a single LM that allows evaluation/inference at all possible level counts up to $Q$, analogous to the quantizer dropout method used to train the codec. To perform CL drop, we first define a *dropout distribution* $\mathcal{P}(q)$ over the all levels $q \in \{1, \dots, Q\}$, and then during LM training, we truncate inputs $\boldsymbol{x}^{(\text{delay})}$ along the level dimension according to $\mathcal{P}(q)$. The LM's training objective can hence be written as:

$$\min_{\theta} \mathbb{E}_{(\boldsymbol{x}, \boldsymbol{y}) \sim \mathcal{D}, Q' \sim \mathcal{P}(q)} \left[ -\log p_\theta \left( \boldsymbol{x}^{(\text{delay})}_{:, :Q'} \mid \boldsymbol{y} \right) \right] \tag{5}$$

where $\mathcal{D}$ is the LM training set with paired conditions $\boldsymbol{y}$ and RVQ codes $\boldsymbol{x}$ for the desired audio, and $\theta$ is the set of the LM's trainable parameters.

For CL drop to be effective in determining the best $Q'$, its end-to-end performance profile across different level counts should closely mirror the trends without CL drop (i.e., the 'end-to-end TTS' curve in Fig. 2). Intuitively, the choice of $\mathcal{P}(q)$ is critical in preserving the trends, as it governs how much the LM's focus is shifted toward the lower (coarser-grained) levels.[4]

---

[4]Experiments on different $\mathcal{P}(q)$'s are deferred to Appendix A.

### 4.3 NAVIGATING OTHER KEY CODEC HYPERPARAMETERS

In addition to the number of RVQ levels ($Q$), there are two additional hyperparameters that affect the compression factor of the codec: (i) the codec's frame duration ($1/f_x$), and (ii) the codebook size ($|\mathcal{V}|$). The bitrate of the codec, equal to $Qf_x \log_2(|\mathcal{V}|)$ bits per second, is a function of these three factors and directly impacts the reconstruction quality. In siloed codec design, these three factors can be traded off freely to optimize for higher reconstruction quality at some fixed bitrate. However, in a co-design context, the LM's behavior can be impacted by different tradeoffs even when the codec's bitrate is kept fixed.

Here we make several observations about frame duration and codebook size respectively in the context of codec-LM co-design. Starting with frame duration, from Eqn. (3), we can observe that the Delay LM training sequence length is inversely proportional to frame duration. Accordingly, increasing it by a factor of two can roughly halve sequence length, resulting in efficiency gains and reduced inference latency. However, codec bitrate is also inversely proportional to frame duration. Therefore, to preserve audio quality with an increased frame duration, we need to also increase either the codebook size $|\mathcal{V}|$ and/or the number of RVQ levels $Q$.

On the other hand, increasing the codebook size $|\mathcal{V}|$ may have mixed impacts on the LM. On the positive side, assuming the frame duration and bitrate are controlled, using a larger codebook (and hence fewer RVQ levels) reduces the extent of packing information from multiple (i.e., $Q'$) code frames into one Delay LM timestep $x_t^{(\text{delay})}$. However, increasing only $|\mathcal{V}|$ while holding $Q$ constant leads to an exponential growth in the LM's vocabulary size (and embedding parameters) relative to a linear increase in bitrate. This growth can inflate the LM's memory footprint and introduce potential modeling challenges. Thus, while our CL drop technique can help determine the best $Q'$ efficiently given a fixed $|\mathcal{V}|$, finding the optimal $|\mathcal{V}|$ still requires trial and error.

All three of the aforementioned hyperparameters—$Q$, $|\mathcal{V}|$, and $f_x$—are relatively straightforward to adjust in open-source codec implementations (Défossez et al., 2023; Kumar et al., 2023). Only changing $f_x$ is achieved indirectly through specifying stride sizes in each strided convolutional layers since $f_s/f_x = \prod_{i=1}^{S} s_i$, where $f_s$ is the audio sampling rate, $S$ is the number of strided convolution layers in the codec encoder, and $s_i$'s are the individual stride sizes.

## 5 EXPERIMENTAL SETUP

**Datasets for codec.** For TTS, we collect 1.7K hours of YouTube podcast data in-house to train the codec. For music experiments, we use the *medium* version of FMA dataset (Defferrard et al., 2017) containing 200 hours of multitrack music. To evaluate audio reconstruction of our codecs, we follow DAC (Kumar et al., 2023) and create a dataset of 3K 10-second audios comprising speech (Mysore, 2014), music (Rafii et al., 2017) and general sounds (Gemmeke et al., 2017) (1K each).

**Datasets for LM.** For TTS, we use the 550-hour LibriTTS-R (Koizumi et al., 2023) for LM training, and its *test-clean* split (8 hours, 4.7K samples) for evaluation. For unconditional music generation, we train our LMs on 1.5K hours of multitrack music from MTG-Jamendo dataset (Bogdanov et al., 2019). We exclude examples with vocals using the associated metadata, and and hold out 1.5K examples for evaluation.

**Codec model specifics.** We utilize the open-source code of DAC (Kumar et al., 2023) and implement our changes on top. Our codecs have 76~84M non-codebook parameters due to various frame durations. We train our codecs for 300K steps with an effective batch size of 75 seconds of audio. We use the AdamW (Loshchilov & Hutter, 2018) optimizer with $10^{-4}$ initial learning rate and exponential decay. The training process takes about 25 hours on 4 NVidia H100 (80G) GPUs.

**LM model specifics.** Following recent validation that a hybrid of state-space model (SSM) and attention outperforms either approach alone (Waleffe et al., 2024; Hatamizadeh & Kautz, 2024), we use 24 layers of stacked Mamba2 (Dao & Gu, 2024) and Transformer decoder blocks (Vaswani et al., 2017), totaling 400M non-embedding parameters. We prepend the conditioning information for TTS (i.e., $\boldsymbol{y}$, which includes text transcripts and speaker embedding) to the RVQ audio codes

Table 1: Codec receptive field settings vs. end-to-end TTS & music generation performance. Our proposed **framewise codec encoder** (Sec. 4.1) consistently beats the default streamable setting (i.e., *Causal*) both on LM likelihood (cf. NLL) and all end-to-end metrics. Stdev over 5 runs follow $\pm$.

| | Framewise? | | Stream? | Codec Recons. | Text-to-Speech | | | | Uncond. Music | |
|---|---|---|---|---|---|---|---|---|---|---|
| Codec | Enc. | Dec. | | Mel-L1↓ | NLL↓ | WER↓ | NISQA↑ | Spk. sim.↑ | NLL↓ | FAD↓ |
| *Non-caus.* | ✗ | ✗ | ✗ | .791 | 5.80 | 3.94 | 4.48 | 82.7 | — | — |
| *All-frame.* | ✓ | ✓ | ✓ | .961 | 4.55 | 3.82 | 3.28 | 70.4 | — | — |
| *Causal* | ✗ | ✗ | ✓ | .846 | $5.46_{\pm.00}$ | $4.12_{\pm.35}$ | $4.35_{\pm.01}$ | $80.2_{\pm.1}$ | $6.06_{\pm.01}$ | $18.7_{\pm1.2}$ |
| **Proposed** | ✓ | ✗ | ✓ | .873 | $\mathbf{4.97}_{\pm.02}$ | $\mathbf{3.71}_{\pm.19}$ | $\mathbf{4.37}_{\pm.02}$ | $\mathbf{80.7}_{\pm.2}$ | $\mathbf{5.16}_{\pm.00}$ | $\mathbf{17.1}_{\pm0.8}$ |

$\boldsymbol{x}^{(\text{delay})}$. The text transcript is transformed into character embeddings, while the speaker embedding is extracted using a raw waveform-based speaker recognition model (Jung et al., 2022).

We train our LMs for 30K steps with a batch size equivalent to 500 seconds of audio. We use the AdamW optimizer (Loshchilov & Hutter, 2018) with a peak learning rate of $4 \times 10^{-4}$, and 10% warmup steps followed by cosine decay. Training takes 12 hours on 8 H100 (80G) GPUs. For inference, we use pure sampling from the LM's output logits.

**Evaluation for audio reconstruction (codec).** We follow (Kumar et al., 2023) and compute the L1 distance between the log-scaled Mel spectrograms of the original and reconstructed waveforms to measure reconstruction at the signal level. We abbreviate this metric as *Mel-L1* hereafter.

**Evaluation for end-to-end audio generation (codec + LM).** To evaluate our end-to-end TTS system involving both the codec and the LM, we consider the following three aspects:

- **Intelligibility:** Following (Wang et al., 2023), we measure the word error rate (WER, in %) between the given text transcript and automatically transcribed text by Whisper (Radford et al., 2023) (v3 large) model from the generated speech.
- **Audio quality:** We leverage NISQA (Mittag et al., 2021) overall quality score, which is predicted by a CNN-Transformer model trained on pairs of speech audios and human-labeled quality scores in the range of $[1, 5]$. NISQA has been shown to correlate well (Pearson's $r \geq 0.9$) with human judgments of speech audio quality.
- **Speaker control:** Following (Wang et al., 2023; Kim et al., 2024), we compute the cosine similarity ($\in [-1, 1]$, reported in %) between the given speaker embedding and that extracted from the generated speech, using the same speaker recognition model (Jung et al., 2022).

For experiments on unconditional music generation, following (Copet et al., 2023; Agostinelli et al., 2023), we report Fréchet audio distance (FAD) (Kilgour et al., 2019) computed on audio embeddings from the VGGish (Hershey et al., 2017) audio classification model. FAD captures how realistic the generations are at the dataset level (i.e., all generations vs. all reference inputs) using feature-wise covariances estimated from all audio embeddings of the generated/reference set.

## 6 RESULTS AND DISCUSSION

We first conduct experiments specifically for each proposed technique (i.e., Sec 4.1, 4.2, and 4.3) to elucidate their individual effects, and finally combine them to show their collective benefits.

**Framewise codec encoder.** Table 1 presents the audio reconstruction and downstream TTS (and music) generation performance, under different receptive field settings for the codec. Following the configurations of DAC (Kumar et al., 2023), we set the frame duration of the codec ($1/f_x$) to 11ms, the number of RVQ levels ($Q$ and $Q'$) to 9, and the per-level codebook size ($|\mathcal{V}|$) to $2^{10}$. The first two rows have clear drawbacks—*Non-caus*, which is default in the official DAC (Kumar et al., 2023), though achieves strongest performance in all aspects, is not low-latency streamable due to dependency on future context. The *all-frame* setting, meanwhile, leads to poor audio quality (NISQA) and speaker control (Spk. sim.), suggesting that framewise receptive fields (see Sec. 4.1)

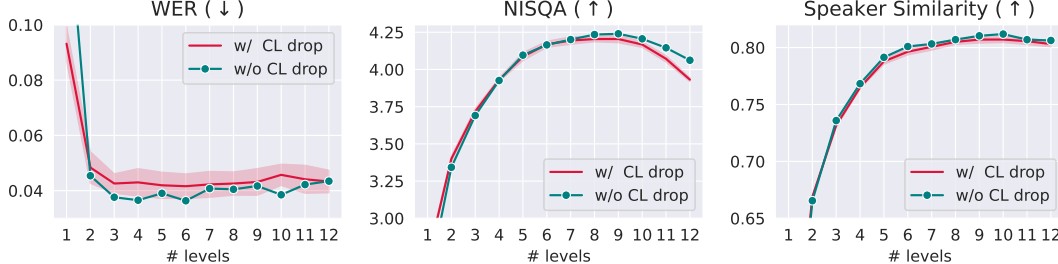

Figure 3: Number of RVQ codebook levels used by LM vs. end-to-end TTS metrics. Training one LM with **codebook level dropout** ('**CL drop**', Sec. 4.2) leads to a performance profile that trends almost identically to training $Q = 12$ LMs without CL drop at each possible level count. Hence, CL drop can help determine the best level count efficiently. Shaded bands represent stdev over 3 runs.

Table 2: Effects of using **longer frame durations** (Sec. 4.3), holding audio reconstruction quality approximately constant by varying codebook size $|\mathcal{V}|$ and/or # of RVQ levels $Q'$. We measure the actual inference time (LM & codec decoding combined) over 50 samples with batch size 1 and treat the first row as the baseline for the 'Inf. speedup' column. In general, using a $2\times$ frame duration (22ms) strikes best balance between performance and efficiency. Stdev over 5 runs follow $\pm$. First row is the default configuration inherited from DAC.

| Codec Config | | | | Codec Recons. | Text-to-Speech | | | Efficiency |
|---|---|---|---|---|---|---|---|---|
| Frame dur. | $\log_2(|\mathcal{V}|)$ | $Q'$ | Rel. bitrate | Mel-L1↓ | WER↓ | NISQA↑ | Spk. sim.↑ | Inf. speedup↑ |
| 11ms | 10 | 9 | $1.00\times$ | .873 | $3.71_{\pm.19}$ | $4.37_{\pm.02}$ | $80.7_{\pm.2}$ | $1.00\times$ |
| 11ms | 15 | 6 | $1.00\times$ | .874 | $3.73_{\pm.28}$ | $4.33_{\pm.01}$ | $80.4_{\pm.1}$ | $1.01\times$ |
| 22ms | 10 | 16 | $0.89\times$ | .888 | $4.21_{\pm.33}$ | $\mathbf{4.42}_{\pm.01}$ | $\mathbf{81.0}_{\pm.1}$ | $1.94\times$ |
| 22ms | 15 | 11 | $0.92\times$ | .876 | $\mathbf{3.55}_{\pm.36}$ | $4.33_{\pm.01}$ | $79.3_{\pm.1}$ | $2.00\times$ |
| 44ms | 10 | 32 | $0.89\times$ | .875 | 6.73 | 4.14 | 76.7 | $3.20\times$ |
| 44ms | 15 | 20 | $0.83\times$ | .871 | 4.53 | 3.65 | 73.2 | $\mathbf{3.77}\times$ |

should not be applied to the codec decoder. Our proposed framewise codec encoder setting (last row) **outperforms** the default streamable *causal* setting consistently, both **on LM likelihood** ($>8\%$ lower NLL) **and all end-to-end TTS and music generation metrics**. Notably, it is slightly worse on Mel-L1 (though relatively close to *causal* compared to the other two settings), underscoring the fact that **better audio reconstruction does not always translate to better end-to-end performance**. Due to its validated advantage, we conduct all subsequent experiments with framewise codec encoders and causal decoders, unless otherwise specified.

**LM codebook level dropout (CL drop).** Results of training the LM with codebook level dropout (see Sec. 4.2) are presented in Fig. 3. To examine how end-to-end performance evolves in the higher-bitrate regime, we use 15-bit codebooks ($\log_2(|\mathcal{V}|) = 15$) and codebook levels $Q = 12$ for the codec, amounting to a maximum bitrate that is $2\times$ that of official DAC. We experiment with various dropout distributions $\mathcal{P}(q)$ (see App. A for details) and conclude that it is best to train at the full level count (i.e., 12 in this case) for 90% of the steps and uniformly distribute the remaining 10% to all lower level counts. The curves in Fig. 3 show that training a single LM with CL drop produces a performance profile closely aligned with training 12 separate LMs without CL drop. This demonstrates that **CL drop is a reliable method for practitioners to efficiently optimize for the level count $Q'$ with significantly reduced training compute**. We also observe that training without CL drop leads to slightly better performance at each particular level count, which is reasonable since the LM does not need to generalize to multiple level counts. In practice, practitioners can train a second LM without CL drop after choosing the best level using the LM trained with CL drop to capture the improvement. Besides, the curves also show that WER, which focuses on (coarser) word-level information, reaches the best early at 3∼4 levels, while NISQA and speaker similarity, which are tied more closely to the fine-grained details, peak at around 9 levels. Since different metrics behave differently w.r.t. level count, determining the best level count is at the discretion of

Table 3: Combined improvements from using multiple proposed techniques—**#1**: Framewise codec encoder; **#2**: CL drop; **#3**: Longer frame duration. $Q'$ denotes the # of levels the LM is trained with for end-to-end TTS, while $Q$ denotes the RVQ codec's full # of levels. We *italicize* the second best setting for each metric. Compared to the baseline using a causal codec (1st row), applying all of our proposed techniques (last row) improves both the efficiency and all end-to-end TTS metrics.

| Proposals | | | Codec Config | | | Text-to-Speech Metrics | | | Efficiency |
|---|---|---|---|---|---|---|---|---|---|
| **#1** | **#2** | **#3** | Frame dur. | $\log_2(\mathvert\mathcal{V}\mathvert)$ | $Q' : Q$ | WER↓ | NISQA↑ | Spk. sim.↑ | Inf. speedup↑ |
| ✗ | ✗ | ✗ | 11ms | 10 | 9 : 9 | 4.12 $_{\pm.35}$ | 4.35 $_{\pm.01}$ | 80.2 $_{\pm.1}$ | 1.00× |
| ✓ | ✗ | ✗ | 11ms | 10 | 9 : 9 | **3.71** $_{\pm.19}$ | 4.37 $_{\pm.02}$ | 80.7 $_{\pm.2}$ | 1.01× |
| ✓ | ✗ | ✓ | 22ms | 10 | 16:16 | 4.21 $_{\pm.33}$ | *4.42* $_{\pm.01}$ | **81.0** $_{\pm.1}$ | *1.95×* |
| ✓ | ✓ | ✓ | 22ms | 10 | 14:16 | *3.86* $_{\pm.19}$ | **4.43** $_{\pm.01}$ | *80.8* $_{\pm.2}$ | **2.01×** |

practitioners depending on use cases. We find that **choosing the best level count based on FAD** (shown in Fig. 2, which uses the same codec as here and would suggest using 9 levels) **achieves a balanced performance between all the TTS metrics** we consider.

**Longer frame duration.** Table 2 displays the effects of using longer frame durations and wider codebooks. Here, we use the number of levels $Q'$ (in this set of experiments, $Q' = Q$) as a variable to roughly control for audio reconstruction quality (i.e., Mel-L1). We also measure the total time of LM inference plus codec decoding under batch size 1 to show the relative inference speed of the different settings (i.e., Inf. speedup). In general, **using a 22ms frame duration** (i.e., 2× the default 11ms for DAC) **preserves or improves TTS performance and enjoys a 2× inference speedup** at the same time. Increasing the frame duration to 44ms leads to substantially worse TTS metrics despite further efficiency gains. However, whether to increase the codebook size $\mathvert\mathcal{V}\mathvert$ from the default $2^{10}$ to accommodate longer frame durations remains unclear (better on WER, worse on other metrics), warranting a more fine-grained exploration (e.g., a dense sweep over 10- to 15-bit codebooks) in future work.

**Combining all techniques.** Table 3 illustrates the cumulative impact of progressively integrating our proposed techniques. The first two rows are derived from Table 1, and the third from Table 2. In the last row, we apply LM codebook level dropout to a (22ms, 10-bit, 16-level) codec, identifying the optimal level count $Q' = 14$ using FAD on end-to-end TTS. Comparing the streamable baseline (1st row) and the final model combining all our techniques (last row), we achieve substantial improvements across all end-to-end TTS metrics, while simultaneously doubling inference speed.

## 7 Conclusion and Future Work

In this work, we presented several co-design techniques for neural codec language models to address the isolation between codec and language modeling research fronts. Specifically, we proposed framewise encoder for the codec and codebook level dropout for the LM, and explored the effects of longer frame durations and wider codebooks. We illustrated the individual benefits of each co-design technique with end-to-end TTS experiments, and demonstrated that applying all of them improves both the performance and efficiency of the codec-LM system.

Future endeavors may (i) study the theory of why framewise compressed representations improve language modeling, (ii) develop RVQ codecs that have flexibility not only in the number of levels, but also in codebook size and frame duration such that our LM codebook level dropout can be applied to all three key hyperparameters altogether, and (iii) uncover the scaling properties (Hoffmann et al., 2022) of the optimal codec settings w.r.t. larger models and more training data.

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

## A  CHOOSING A GOOD $\mathcal{P}(q)$ FOR LM CODEBOOK LEVEL DROPOUT

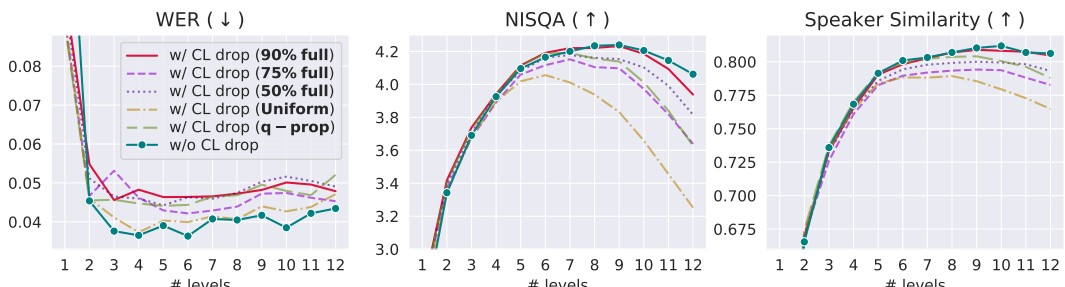

Figure 4: Effects of using different dropout distributions, i.e., $\mathcal{P}(q)$, for LM codebook level dropout. The curves of 'w/ CL drop' settings are the closer to those of 'w/o CL drop' the better.

For LM codebook level dropout (i.e., CL drop) to be effective in determining the optimal level count, its performance profile w.r.t. the level count should trend as closely as possible to that resulting from training LMs without CL drop at every possible number of levels. Here, we find that the choice of dropout distribution $\mathcal{P}(q)$, which determines the fraction of training steps allocated to each level count, to be critical. We experiment with a total of 5 different $\mathcal{P}(q)$'s detailed below:

- **Uniform:** $\mathcal{P}(q) := \frac{1}{Q}$; $\forall q \in \{1, \ldots, Q\}$, i.e., every level count gets equal attention.

- $q$-**proportional (or $q$-*prop*):** $\mathcal{P}(q) := \frac{q}{Z(Q)}$; $\forall q \in \{1, \ldots, Q\}$, where the normalization constant $Z(Q) := \sum_{q'=1}^{Q} q'$, i.e., the fraction for each level count $q$ is proportional to $q$.

- **50% full:** $\mathcal{P}(q) := 0.5$ for $q = Q$, and $\mathcal{P}(q) := \frac{1-0.5}{Q-1}$; $\forall q \in \{1, \ldots, Q-1\}$, i.e., the full level count $Q$ gets 50% of the steps, and all the lower level counts share the remaining 50% uniformly.

- **75% full:** $\mathcal{P}(q) := 0.75$ for $q = Q$, and $\mathcal{P}(q) := \frac{1-0.75}{Q-1}$; $\forall q \in \{1, \ldots, Q-1\}$, which is similar to **50% full** but focuses more on the full level count $Q$.

- **90% full:** $\mathcal{P}(q) := 0.9$ for $q = Q$, and $\mathcal{P}(q) := \frac{1-0.9}{Q-1}$; $\forall q \in \{1, \ldots, Q-1\}$, which puts even more focus on $q = Q$ than **75% full**.

The performance profiles resulting from these $\mathcal{P}(q)$'s are shown in Fig. 4. The NISQA (which evalutes *audio quality*) and speaker similarity profiles suggest that **90% full** is the best choice among the five $\mathcal{P}(q)$'s. Other choices all peak at relatively lower level counts, and **Uniform**, which is the most straightforward option, appears to be the worst of the five.

The reasons behind why allocating only 10% to lower level counts leads to metrics that track most closely those from training separate LMs for each level count are left for further investigation. Our intuition is that, training with $Q$ levels already includes modeling all the lower levels, and hence the LM only needs a small number of steps to adapt to the scenarios where the finer-grained information in higher levels is absent.

