# OpenReview forum: "Towards Codec-LM Co-design for Neural Codec Language Models"
_ICLR.cc/2025/Conference — ICLR 2025 Conference Withdrawn Submission_

### Official Review · Reviewer_ojP5 · 2024-10-29

**Soundness:** 3
**Presentation:** 3
**Contribution:** 3
**Rating:** 6
**Confidence:** 4

**Summary:**

The paper investigates the audio codec from the viewpoint of codec-LM-based TTS. The authors provide several interesting insights, including (1) the proposal of limiting the receptive field for the codec model to improve TTS quality while degrading codec quality, (2) the insight that the best number of RVQ layers used for TTS can be inferred based on a model trained with codebook dropout during codec-LM training, making the parameter sweep efficient, and (3) experimentation using different token rates and codebook sizes in the same or similar bps configurations.

**Strengths:**

The contributions in (1) and (2) are specifically interesting in the sense that (1) is counter-intuitive and (2) sheds light on a point that was previously missing in the field. The paper is very clearly written, with all results appearing valid.

**Weaknesses:**

This is purely an investigation paper without theoretical or architectural advancements. I also think that the observation from point (3) is within expectation and does not bring much impact to the community. While point (1) can be regarded as an architectural improvement, more exploration would be required to be fully convincing.
  - The framewise encoder fundamentally limits the codec quality and sets an upper bound on the quality achievable by the codec-based TTS model. Then, could it be possible that better training of codec-LM (better hyperparameters, better model architecture, or just more training compute) could reverse the observation that "framewise encoder is better"? If so, could you add further discussion on this concern?
  - Please include WER, NISQA, and Spk Sim for codec reconstruction in Tables 1 and 2.
  - Please add standard deviations for all rows in Tables 1 and 2. It is odd to include them only occasionally.
  - For completeness, I’d suggest adding the setting where the encoder is not framewise, and the decoder is framewise for Table 1. Without this, the statement in Section 6, “The all-frame setting, … suggesting that framewise receptive fields (see Sec. 4.1) should not be applied to the codec decoder” is not fully supported.

Minor comment:
  - It is not appropriate to report speaker similarity in percentages because it is not a notion ranging from 0 to 100. You can simply report the number as is (similar to Mel-L1).

**Questions:**

Please address the questions raised in the weaknesses section.

---

### Official Review · Reviewer_kau5 · 2024-10-30

**Soundness:** 2
**Presentation:** 3
**Contribution:** 2
**Rating:** 5
**Confidence:** 5

**Summary:**

This work explores how an acoustic codec can be adapted for language model training and introduces three core techniques. However, I believe these are more akin to design "tricks".
The first technique involves ensuring non-overlapping receptive fields in the encoder of codec. The authors argue that overlapping fields "may muddy the relationship between time in the waveform and time in the code frames," but this lacks a detailed theoretical explanation.
The second technique discusses the relationship between token per second (longer frame durations) and codebook size, though it seems less insightful compared to recent concurrent work, such as WavTokenizer. The third technique examines the connection between the number of quantizers and text-to-speech (TTS) performance, introducing "CL DROP" to avoid repeated LLM training.
However, from an engineering perspective, this seems more like common practice.

**Strengths:**

1. In terms of motivation, the use of independent receptive fields in the encoder may be an interesting trick, but it lacks a solid theoretical justification and requires more robust experimental validation.
2. The finding regarding longer frame durations (similar to the concurrent work on WavTokenizer) is noteworthy; however, it necessitates a more detailed hyperparameter analysis and a more thorough evaluation of the experimental results.
3. While the authors present several experimental analyses, such as in Figure 2, I believe this insight has long been considered common knowledge in the industry.

**Weaknesses:**

From the motivational perspective:
1. the theoretical foundation underlying the independence of the features in the encoder is notably lacking. The authors must provide a rigorous explanation for this, as emphasized in the limitations section. This deficiency also raises two potential issues: first, there is a concern that the model may lose its ability to capture contextual information, leading to semantic omissions in the codec; second, the complete independence of the receptive fields may introduce boundary issues when reconstructing relevant textual information.
2. the rationale for using "longer frame durations" and the choice of a codebook size(32768) of warrant clarification. If the codebook is excessively large, it may result in decreased utilization and, consequently, adversely affect TTS performance.
3. The LM codebook level dropout appears more as an engineering experiment with limited applicability, particularly since architectures like VALL-E and SoundStorm utilize random selections of quantizers during training.
4. The innovations presented in these three tricks are not particularly groundbreaking within the codec domain. From the CodecLLM perspective, attention might be better directed toward bridging the gap between reconstruction paradigms and LLM generation paradigms, as well as investigating the impact of integrating acoustic tokens on the degradation of text-based LLM performance. These three tricks in this paper appear to be unrelated to the broader objectives(and considering the substantial existing work on codec compressibility).

From the experimental perspective:
1. the rationale for employing internal datasets among numerous open-source options raises questions about the potential biases introduced by dataset selection.
2. Additionally, both codec and TTS evaluations lack a number of relevant metrics; for instance, the codec assessment should include PESQ, VISQAL, UTMOS, and STOI results, while TTS evaluations should incorporate MOS metrics.
3. Based on Table 1, the effectiveness of the tricks employed in this paper appears to be modest, with no significant differences observed in WER, NISQA, and Spk metrics for TTS tasks. The codec's reconstruction quality also does not demonstrate substantial improvements.
4. Figure 3 indicates that CL drop does not enhance performance and may be influenced by training biases.
5. There is the need for an analysis of codebook utilization similar to that found in the WavTokenizer paper in Table 2, particularly concerning frame duration and codebook space design.
6. As plug-and-play tricks, it is essential to validate its effectiveness across various codec models or assess its adaptability within different codec TTS architectures (e.g., diffusion models, VALL-E). Alternatively, the introduction of additional baselines could be beneficial.

In conclusion, I eagerly anticipate witnessing innovative paradigm designs in the acoustic codec domain (e.g., Encodec, SpeechTokenizer) or models that deliver remarkable results (e.g., DAC). However, this current work still exhibits certain gaps that need to be addressed.

**Questions:**

1. "they focused on single, and highly specific, modifications to the codec. Meanwhile, our work investigate the downstream impact of multiple general RVQ codec hyperparameters in combination, painting a more complete picture for end-to-end system practitioners.", The authors claim that the main distinction between their work and previous studies lies in the combination of various tricks?
2. "the frame duration and codebook size. In siloed codec design, these hyperparameters can be freely traded off with one another without affecting the codec's bitrate or audio reconstruction quality", Why do the frame duration and codebook size not impact reconstruction performance?
3. Could the authors clarify whether they will fully open-source all weights, as well as the complete code for training and inference

---

### Official Review · Reviewer_aYvC · 2024-11-04

**Soundness:** 4
**Presentation:** 3
**Contribution:** 3
**Rating:** 6
**Confidence:** 5

**Summary:**

This paper addresses a critical gap in the integration of neural codec design and codec language modeling, particularly in the context of TTS and audio generation tasks. The authors argue that while both codecs and LMs are essential components in audio generation systems, they have traditionally been developed in isolation, limiting their potential for synergy. To bridge this gap, the authors propose three experimental co-design strategies: a frame-wise codec encoder, LM codebook level dropout, and increased codec frame duration. These strategies aim to enhance the efficiency and performance of end-to-end TTS systems by optimizing the interactions between codec and LM components.

**Strengths:**

1. The point of view of this paper - Codec LM and Codec co-design - is very meaningful. This paper gives consideration to both efficiency and effect, which is of guiding significance to the design of Codec LM and Codec.
2. Writing is good. The paper is well-structured and articulates complex ideas clearly, besides, the experiment can well support the claim of the paper.

**Weaknesses:**

1. Current Codec training also uses layer drop, similar to the CL Drop proposed by the authors. Perhaps the use of layer drop during Codec training will also have an impact on the effectiveness of Codec LM. The author could explore this further.
2. The trick of increased codec frame duration is not novel, and it is just an engineering trade-off. In my opinion, the efficiency comparison in the paper should not consider different frameshift, because this is a definite efficiency improvement.
3. The entry point and approach make sense, however, my main concern is that there is no apple-to-apple comparison. The paper uses stacked Mamba2 and Transformer blocks, instead of existing TTS architecture such as VALL-E or AudioLM. In addition, the evaluation of audio quality did not report the results of PESQ or VISQOL. I hope to see the efficiency and effectiveness compared with existing works under the same model architecture with these tricks.

**Questions:**

See above.

---

### Official Review · Reviewer_ZVn1 · 2024-11-04

**Soundness:** 2
**Presentation:** 3
**Contribution:** 2
**Rating:** 3
**Confidence:** 3

**Summary:**

This paper presents co-design strategies to integrate codecs and language models in text-to-speech systems, introducing a framewise codec encoder, LM codebook level dropout, and extended frame durations. These enhancements double inference speed and improve intelligibility, audio quality, and speaker control. The work highlights significant efficiency and performance gains, with plans for open-sourcing implementations and exploring further scalability and flexibility in future research.

**Strengths:**

1. The paper bridges the previously separate fields of codec and LM design with practical co-design strategies that require no major architectural changes, offering immediate applicability, especially in TTS systems.
2. The introduction of a framewise codec encoder enhances both LM performance and TTS metrics, while the LM codebook level dropout efficiently resolves the costly hyperparameter search. Additionally, frame duration optimization effectively improves inference speed without sacrificing quality.
3. The paper effectively validates its proposals, showing that integrating the techniques enhances language model performance, supported by thorough quantitative and qualitative analyses.

**Weaknesses:**

1. The techniques primarily focus on the multi-codebook tokenizer structure in RVQ. However, their effectiveness, particularly when employing the framewise codec encoder and longer frame duration approaches, requires further validation when applied to single-codebook speech tokenizers in combination with LMs.
2. The use of LM codebook level dropout (CL drop) and longer frame duration are common techniques in both speech LMs \[1,2] and codec models. Presenting them as key contributions may lack sufficient innovation.
3. The paper's Table 3 presents only four permutations of the three techniques. The remaining two combinations should also be explored and reported to provide a comprehensive understanding of the effects.
4. The experiments are predominantly focused on TTS applications, limiting insight into how the techniques perform in other audio generation tasks. Additionally, these experiments are mostly conducted on a single codec architecture variant, potentially restricting the generality of the findings.

[1]:Wang C, Chen S, Wu Y, et al. Neural codec language models are zero-shot text to speech synthesizers[J]. arXiv preprint arXiv:2301.02111, 2023.

[2]:Borsos Z, Sharifi M, Vincent D, et al. Soundstorm: Efficient parallel audio generation[J]. arXiv preprint arXiv:2305.09636, 2023.

**Questions:**

See details in Paper Weaknesses

---

### Note · Authors · 2024-11-19

**Comment:**

Dear all reviewers,


We sincerely thank all the reviewers for dedicating your time and effort to thoroughly review our manuscript and provide thoughtful, constructive feedback. We greatly appreciate your recognition of our co-design focus as meaningful, the results on frame-wise codes as interesting, and our manuscript as well-written.


We fully agree that the key areas for improvement are:
1. providing theoretical insights and justifications on why frame-wise codes helps downstream language modeling, and
2. demonstrating broader applicability of our findings across tasks, as well as various codec and sequence model variants.


Given the scope of these improvements and the limited rebuttal window, we have decided to withdraw our current submission. We greatly value your feedback, which will be instrumental as we revisit and refine this project in the future.


Thank you once again for your time and insightful feedback!


Sincerely,
Authors of Submission 5407

**Withdrawal Confirmation:**

I have read and agree with the venue's withdrawal policy on behalf of myself and my co-authors.